# Is it best to add native shrubs to a coastal sage scrub restoration project as seeds or as seedlings?

Kylie D. F. McGuire[1]*, Katharina T. Schmidt[1], Priscilla Ta[1], Jennifer J. Long[1], Matthew Yurko[2], Sarah Kimball[1]

**1** Center for Environmental Biology, University of California, Irvine, California, United States of America,
**2** Project Grow (California Coastal Commission), Back Bay Science Center, Newport Beach, CA, United States of America

* kdmcguir@uci.edu

**Data Availability Statement:** The data underlying this study are available at the Dryad data repository; the DOI is 10.5061/dryad.95x69p8jm.

## Abstract

Ecological restoration frequently involves the addition of native plants, but the effectiveness (in terms of plant growth, plant survival, and cost) of using seeds versus container plants has not been studied in many plant communities. It is also not known if plant success would vary by species or based on functional traits. To answer these questions, we added several shrub species to a coastal sage scrub restoration site as seeds or as seedlings in a randomized block design. We measured percent cover, density, species richness, size, survival, and costs. Over the two years of the study, shrubs added to the site as seeds grew more and continued to have greater density than plants added from containers. Seeded plots also had greater native species richness than planted plots. However, shrubs from containers had higher survival rates, and percent cover was comparable between the planted and seeded treatments. Responses varied by species depending on functional traits, with deep-rooted evergreen species establishing better from container plants. Our cost analysis showed that it is more expensive to use container plants than seed, with most of the costs attributed to labor and supplies needed to grow plants. Our measurements of shrub density, survival, species richness, and growth in two years in our experimental plots lead us to conclude that coastal sage scrub restoration with seeds is optimal for increasing density and species richness with limited funds, yet the addition of some species from container plants may be necessary if key species are desired as part of the project objectives.

## Introduction

In the light of widespread extinctions caused by global climate change, the ecological restoration of degraded plant communities is critical to promoting healthy habitats and maintaining biodiversity [1–4]. One key challenge to conducting successful restoration is the identification of optimal approaches, especially given limited funds [5–7]. Heavily degraded areas frequently have a depleted native seed bank, so restoration practices include adding natives from seed

**Funding:** This study was supported by the Center for Environmental Biology (CEB) at the University of California, Irvine (UCI) and by Project Grow, a non-profit project of the California Coastal Commission and the Tides Center.

**Competing interests:** The authors have declared that no competing interests exist.

and transplanting native seedlings [8]. Although there is a large difference in the cost of restoring from seeds or seedlings [5], the overall effectiveness is not documented for many plant communities. In our Southern California coastal sage scrub study system, similar to other invaded communities, non-native species compete with natives for space, water, and nutrients [9–11].

Using seeds for restoration can be advantageous because seeds are less expensive than container plants [12]. Seeding may also be advantageous because it requires less time since additional preparatory steps typically associated with growing container plants would be avoided [13,14]. Additionally, using seeds may be the only option for some sites, such as those with steep slopes [15]. Restoring from seeds can also be challenging for several reasons, including low germination rates, defining seeding rates, increased potential for competition with non-natives during vulnerable early growth, and identifying optimal seeding techniques [16–18]. Many non-native invasive species have early germination and growth compared to natives, giving them a competitive advantage [19,20]. Adding native plants to the restoration site as seeds means that the seeds may be out-competed early in the growing season, during seedling establishment [21]. Defining seeding rates is of high importance when restoring from seeds [22,23]. With rates that are too low, the project will see low recruitment. With rates that are too high, funds and resources go to waste [24,25]. Seeding methods must also be considered since seeds of different species have varied success depending on how they are sown [26].

Practitioners may decide to use container plants to overcome some of the challenges of restoration from seeds. Seeds are frequently limited in restoration, and challenges such as seed storage, dormancy, and viability make it difficult to ensure restoration success [27,28]. It may be considered safer to optimize available seed by growing out plants and adding them to the site as seedlings [29]. Container plants will also have a head start on the invasives [30], which contributes to the fact that plants transplanted from containers tend to have higher survivorship in restoration projects than plants added by direct seeding [12]. However, restoring from container plants poses its own challenges. For one, using container plants is generally much more expensive than purchasing seeds [13], which places limits on the quality of restoration that can be achieved with a given budget. Using container plants may also be more time-consuming considering the extra labor involved in growing and tending to the young plants [31]. Although this work can be outsourced to nurseries, this would understandably increase the cost of using container plants. Container-grown plants may also face physiological disadvantages, such as poor root development or root-boundedness, that make the successful establishment more difficult and have lasting effects on plant development [32,33]. It may not be feasible to use container plants in more remote locations due to difficulties with transportation and with irrigation [34].

In this study, we aim to understand the most effective way to add native shrub species to a coastal sage scrub restoration site—as seed or as container plants. Specifically, we hope to answer the following questions: (1) Which is the more successful restoration technique for: a) increasing the percent cover, richness, and density of native plants?; and b) increasing survivorship and growth of native shrubs?; (2) Which restoration technique costs the least money or hours of labor? (3) Are the results consistent or do they vary by species and by functional traits? We hypothesized that container plants would be best for increasing cover because they are larger when added to the site, while seeds are generally better for increasing richness because they allow for annuals to be added. We hypothesized that seeded plots would have more unique species growing in a standardized area due to smaller stature of seedlings that germinated in the field. Plants grown from seed might grow faster and have higher survivorship due to improved root health [32,33]. We also hypothesized that restoration from seed would be less expensive and time-consuming than restoration from container plants [5].

Finally, we did not have any reason to expect the plants' success to vary significantly by species or functional traits.

## Materials and methods

### Site description

The study site is located on a north-facing slope in Newport Valley, part of the Newporter North Environmental Study Area in the city of Newport Beach, California (33°37'10.86"N 117°53'16.39"W;). The Newport Valley consists of a mixture of native and non-native species, with the south-facing slope dominated by native shrubs that were established through previous restoration efforts, gradually transitioning to riparian vegetation at the valley bottom. The pre-restoration vegetation of the north-facing slope, where our study is located, was dominated by non-native species, including *Hirschfeldia incana*, *Brassica nigra*, *Urtica dioicia*, and *Sonchus oleraceus*. The few scattered native species included the shrubs *Artemisia californica* and *Baccharis pilularis*. The climate is Mediterranean, with mild, wet winters and hot, dry summers [35]. The mean annual precipitation is 317 mm with rainfall occurring mainly from November to April (NWS station E3141; Fig 1).

### Experimental design

Our study spanned over two years from October 2016 to September 2018. In early 2016, seven replicate blocks (6.5 m x 12.5 m) were established across our study site to evaluate the differences in restoration success of native plant addition through seeds or container plants (S1 Fig). Each block was enclosed with chicken wire fence to prevent herbivory, due to a large rabbit population at the site. Each block contained two shrub plots (3 m x 5 m), four grass plots (1 m x 5 m), and two forb plots (1 m x 6 m). To determine if the response to seeding versus planting

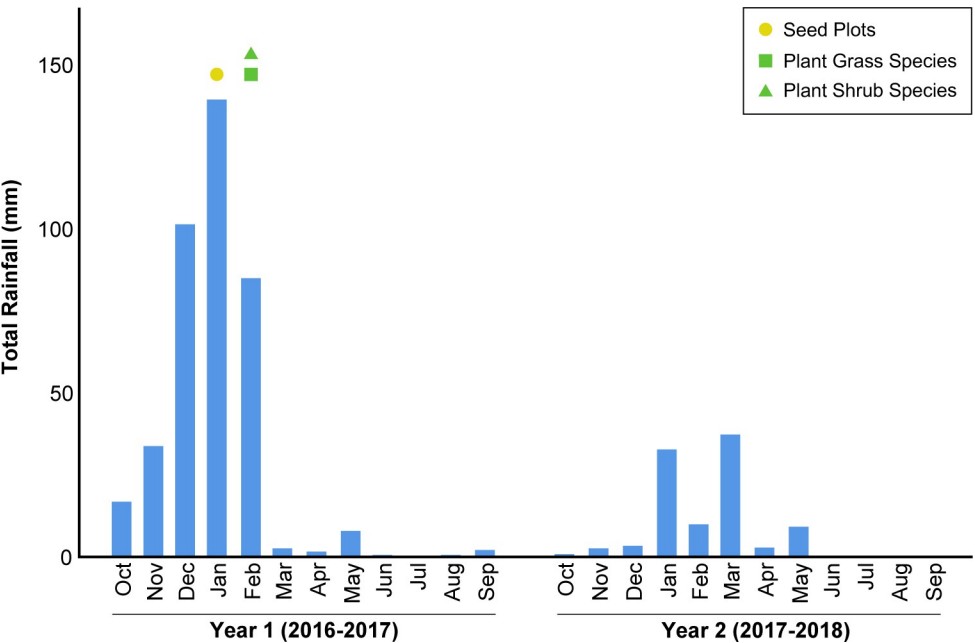

**Fig 1. Monthly rainfall totals for each year of the study.** In addition to monthly rainfall totals for each year of the study, seeding and planting events are also noted on the month that the event occurred. Data are from the Corona del Mar Weather Station (NWS station E3141).

depended on the species of shrub used, twelve native shrub species were included in each shrub plot. Of the four grass plots, *Stipa pulchra* was added to two plots. *Stipa lepida* was added to the other two. Grass and shrub plots were randomly assigned to a seeded or planted treatment. Because annual forbs are rarely, if ever, added to restoration sites as plants, we did not study the relative success of planted and seeded forbs. Instead, all forbs were added to the site as seeds, with the intent of increasing native plant diversity and creating a healthier restored ecosystem than could have been achieved by using shrubs and grasses alone (S1 Fig).

## Seeding and planting

The site received varying amounts of rainfall each growing season (from November to April), with the first (2016–2017) year receiving a mean annual amount of 605 mm while the second (2017–2018) year received 145 mm rainfall. We seeded and planted in late January and early February (Fig 1). Seeds were purchased through S&S Seeds Inc. (Carpinteria, CA). Seeds of some species (*Acmispon glaber*, *Eschscholzia californica*, *Lupinus bicolor*, *Lupinus succulentus*, *Malosma laurina*, *Penstemon spectabilis*, *Phacelia cicutaria*, and *Rhus integrifolia*) were subjected to dormancy-breaking techniques prior to seeding (S1 Table). Seeds of all shrub, grass, and forb species were combined into their respective seed mixes. The seeding rates varied greatly between different species, as is common in coastal sage scrub restoration [36,37]. Due to previous difficulties with restoring the site, we chose to use higher seeding rates than are typical. The seeding rate for *Acmispon glaber* is unusually high due in part to a calculation error, in which the intended seeding rate was doubled.

In January 2017, all experimental plots were seeded and planted. We used hand-broadcasting, raking, and tamping, which has been shown to yield greater germination in coastal sage scrub communities compared to other methods [38]. Seed mixes were sprinkled onto the ground by hand, using no specific spatial pattern, while taking care to distribute the seed evenly throughout each plot.

Native plants were grown from the same S&S Seeds stock as was used for seeding. During the first year of the study, we grew plants in the UC Irvine greenhouse. Seeds were grown in flats, transplanted into 5-cm diameter pots, and finally 10-cm diameter pots once they had reached sufficient size. The container plants were then transferred to the outdoor nursery at Back Bay Science Center (adjacent to the Upper Newport Bay) to allow the plants to gradually be exposed to more extreme environmental conditions before being added to the study site.

In late January of 2017, two individuals of each of the twelve shrub species were added to each of the seven planted-treatment shrub plots, for a total of 14 planted individuals per shrub species (S1 Table). Between 47 and 69 *Stipa* individuals were added to each planted-treatment grass plot. We spaced out all the plants within their plot boundaries to decrease competition and planted natives slightly lower than ground level, building a small soil "moat" around the plant for better water retention. All blocks at the site were hand-weeded throughout the two years of the experiment to remove any non-natives that germinated at the site, with maintenance events occurring approximately twice a month during the winter and spring months.

## Data collection

Although our initial intent was to compare seeded and planted treatments for both shrubs and grasses, we observed very low germination and survival of all grasses. Because of this, we focused our data collection efforts on the shrub plots (S2 Fig).

To compare differences in restoration success when natives are added as seeds or as plants, we collected data on shrub size, density (total number of native plants per 0.25 m$^2$), species richness (number of native species per quadrat), and percent cover of vegetation. Plant density

data were collected in April and May of each year. We randomly placed a 50 cm by 50 cm quadrat near the center of each plot and counted all native and non-native species rooted within the quadrat. Percent cover data were collected once, in March 2018. We installed five 4 m vegetation sampling transects within each of the 3 m x 5 m shrub plots. Transects were spaced 0.5 m apart from each other with point data collected every 1 m to reach a total of 25 points per plot. At each point, we recorded all species present.

To evaluate the differences in overall plant health between seeded and planted individuals, we flagged individuals and recorded survivorship over time. In May and July, we monitored a total of 28 individuals per shrub species, with 14 replicate individuals per treatment, flagging two individuals per each of the seven replicate experimental blocks. If not enough individuals existed for a certain species within a block, extra individuals of that same species were flagged in other blocks. For the planted-treatment plots, we flagged and monitored the individuals we planted. For the seeded-treatment plots, we selected the healthiest looking individuals to monitor over time. Because of the invaded nature of the site, we assumed that shrub germinants were from the seeds that we added and not from natural regeneration.

We collected plant size measurements (plant height and width at the widest point) on flagged native shrubs to compare differences in shrub volume and growth. Shrubs were checked on several different dates (5/30/17, 7/27/17, 10/27/17, 5/19/18, and 5/26/18) and either measured or noted as dead. We used these data to calculate the average lifespan (the number of days between the estimated date of germination and the date of measurement). We used size measurements from 5/26/18 to compare final size, and the change in size from 5/30/17 to 5/26/18 to compare growth. Relative growth was calculated as relative change in size (height or width) using the formula: (size on 5/26/18 − size on 5/30/17)/size on 5/30/17.

## Cost comparison

We tracked hours of labor during the first year of our study (2016–2017) to compare differences in time devoted to seeding or planting. These tasks included plant grow-out, planting, seed mix preparation, seeding, and site preparation and maintenance, with more detailed descriptions provided in S2 Table. We compared the total number of hours associated with seeding or planting to determine which method cost the least amount of time.

We also compared the total amount of money spent on materials for seeding or planting efforts during the first year of our study to determine which technique was the least expensive. The specific materials purchased are stated in S2 Table. Since forb species were always sown directly on-site to mimic local restoration methods, we did not include the cost of restoring forbs in our analysis because there would be no difference. Labor is a major expense associated with any restoration effort, so we also included the monetary cost of labor in our evaluation of the cheapest restoration technique [6,39–41]. We calculated the cost of labor assuming an hourly wage of $19.33 and using the total amount of time spent on preparations, seeding, planting, and site maintenance [42]. While we did not use nursery-grown container plants in our study, we included this cost in our evaluation because nursery-grown container plants are often purchased and used in local restoration efforts.

## Data analysis

All data analysis was performed using R unless otherwise noted [43]. Due to unusually high weed invasion, block 5 was excluded from all analyses. We used the nmle package in R to run mixed-model ANOVAs with treatment as the fixed factor and experimental block as the random factor to determine whether the percent cover of native and non-native plants varied by treatment. Data on the density of native and non-native plants as well as on species richness

was collected at four time points (April and May in both years). To determine whether overall native and non-native density and native species richness were influenced by date of measurement, by treatment, or by the interaction between date and treatment, we performed repeated-measures ANOVAs using SAS proc mixed, with the plot as the repeated factor, date of measurement, treatment, and the date-by-treatment interaction as fixed factors, and block as a random factor [44]. For the density data, final height and width of native shrubs, and the relative change in height and width of native shrubs, we ran separate models that also included species as a fixed factor to test whether species responded differently to the restoration treatments. For these analyses, we included treatment and the species-by-treatment interaction as fixed factors, with block as a random factor. *Atriplex lentiformis* had only one surviving shrub in the planted treatment, while *Elymus condensatus*, *Isocoma menziesii*, and *Rhus integrifolia* had no individuals in the seeded treatment, so those species were excluded from the analysis that included species as a factor. Density was averaged over time for the analysis that included species as a factor after we tested for and found no significant species-by-time interaction. We used logistic regression with a logit link in SAS to evaluate whether the probability of survival varied depending on treatment, species, or the treatment-by-species interaction.

Shrub functional traits (leaf lifespan, flowering time, root depth, and post-fire strategy) were determined by a combination of literature review and our own observations [45–82].

## Results

Restoration conducted by adding seeds and by planting container plants successfully increased native cover from less than 1% before restoration to an average of 50% to 100% by the end of the first year of our study (Figs 2 and S3). Nearly all of the increase in native cover resulted from plants added to the site in the first year of our study, plus a number of "volunteer" natives that germinated from the seed bank following non-native removal (S3 Fig). We monitored plants that successfully established in the first year of the study during the second, dry year as well, tracking changes in density, height, width, and survivorship. There was extremely low germination for both species of perennial bunchgrasses (*Stipa pulchra* and *Stipa lepida*) and low survival of both *Stipa* species from container plants (S2 Fig). All native forbs were added to the site as seeds, consistent with common practices for restoration of this system, and they germinated well, greatly increasing the diversity and density of natives at the site (blue bars in S2 Fig and S1 Table). All results presented below are from the shrub plots, monitored throughout the first and second years of the study.

### Percent cover, density, and richness of native plants

The total percent cover of seeded shrubs was marginally greater than the total percent cover of planted shrubs (Tables 1 and 2, Fig 2A). There was no significant effect of treatment on the cover of non-native species. When the species of the native shrub was included as a fixed factor in the analysis of percent cover data, there was a significant interaction between species and treatment (Table 1, Fig 3A). *Acmispon glaber*, *Atriplex lentiformis*, *Encelia californica*, *Eriogonum fasciculatum*, and *Salvia mellifera* had greater cover when seeded. *Elymus condensatus*, *Isocoma menziesii*, *Malosma laurina*, and *Rhus integrifolia* had greater cover when planted, but very low cover in both treatments. Cover of *Artemisia californica*, *Baccharis emoryi*, and *Peritoma arborea* showed no effect of treatment.

The density of native shrubs was significantly greater in the seeded treatment than in the planted treatment (Figs 2B and 4A). The density of plants decreased through time such that the effect of time of measurement was also significant, and there was a significant treatment-by-year interaction. Most species had similar densities in the seeded and planted treatments.

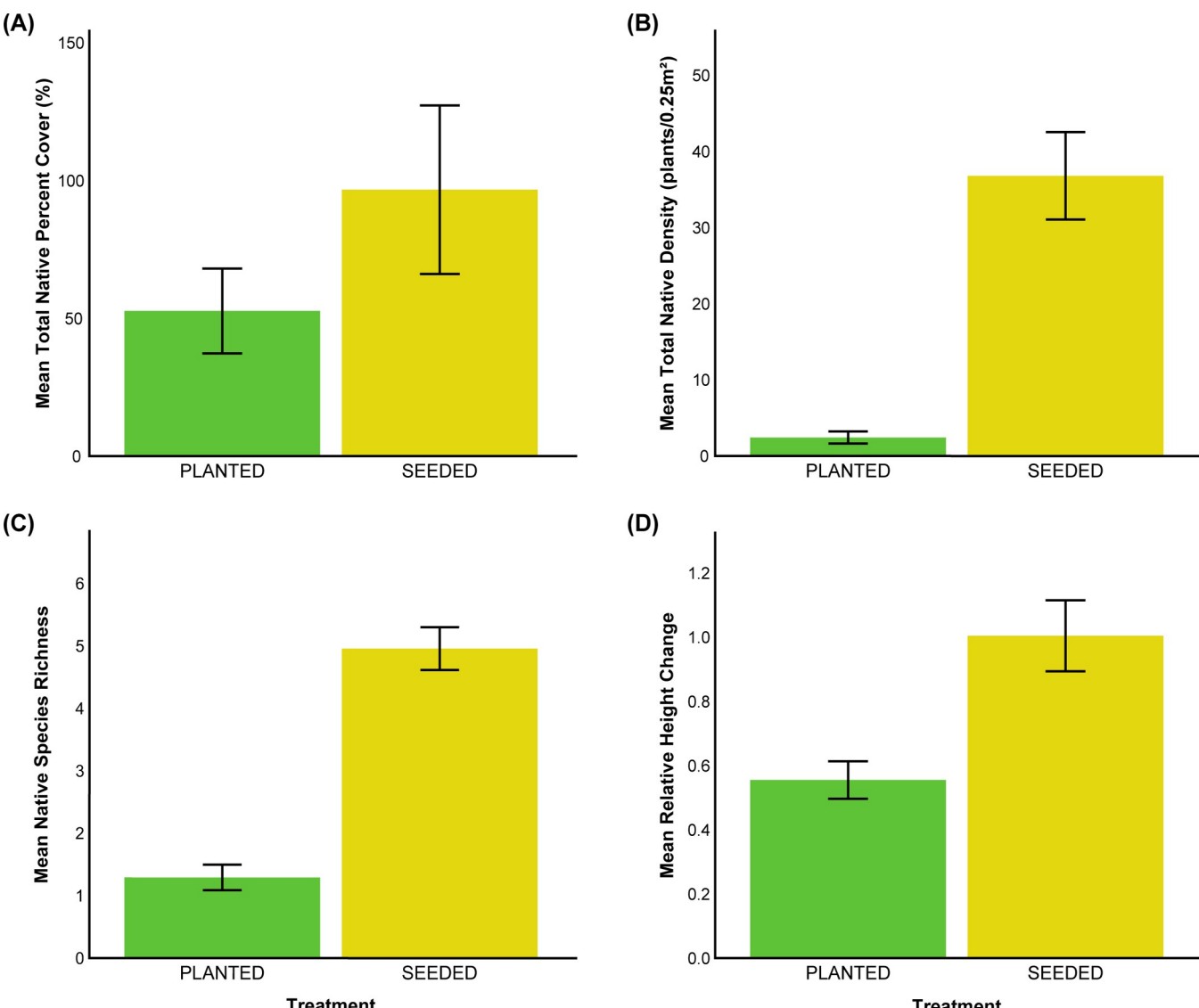

**Fig 2. Average values by treatment for all native shrubs added to the site.** Average plot values for all native shrubs added to the site including A) total native percent cover, B) total native density, C) native species richness, and D) relative change in height. Bars represent mean +/− 1 SE.

The exceptions are *Salvia mellifera*, *Artemisia californica*, and *Atriplex lentiformis* which had somewhat higher density in the seeded treatment, and *Acmispon glaber*, which had significantly higher density in the seeded treatment.

The seeded plots contained significantly higher native species richness than the planted plots (Fig 4B). There were more species in April of the first year of the study than in the later measurement dates and the treatment-by-year interaction was not significant, such that seeded plots continued to contain a greater richness of native shrubs compared to planted plots.

## Growth, lifespan, and survival of native plants

When we compared relative change in height, shrubs in the seeded treatment tended to grow more than those in the planted treatment (Fig 3E). There was no significant effect of species on relative change in height nor was there a significant species-by-treatment interaction. Analysis

**Table 1. Results of statistical analyses.**

| ANOVA | | | | |
|---|---|---|---|---|
| | | **DF** | **F** | **P** |
| **Cover of Native Shrubs** | Treatment | 1, 115 | 3.388 | 0.068 |
| | Species | 11, 115 | 7.688 | **<0.0001** |
| | Treatment:Species | 11, 115 | 3.074 | **0.001** |
| **Total Native Shrub Cover** | Treatment | 1, 5 | 5.952 | 0.059 |
| **Total Non-Native Cover** | Treatment | 1, 5 | 0.508 | 0.508 |
| **Relative Height Change** | Treatment | 1, 129 | 7.310 | **0.008** |
| | Species | 7, 129 | 0.916 | 0.496 |
| | Treatment:Species | 7, 129 | 0.925 | 0.490 |
| **Relative Width Change** | Treatment | 1, 129 | 1.257 | 0.264 |
| | Species | 7, 129 | 3.739 | **0.001** |
| | Treatment:Species | 7, 129 | 1.425 | 0.201 |
| **Lifespan** | Treatment | 1, 201 | 9.970 | **0.002** |
| | Species | 8, 201 | 6.849 | **<0.0001** |
| | Treatment:Species | 8, 201 | 4.303 | **0.0001** |
| **Repeated-Measures ANOVA** | | | | |
| | | **DF** | **F** | **P** |
| **Species Richness*** | Treatment | 1, 35 | 144.67 | **<0.0001** |
| | Date | 3, 35 | 4.54 | 0.009 |
| | Treatment:Date | 3, 35 | 1.51 | 0.23 |
| **Density*** | Treatment | 1, 35 | 124.22 | **<0.0001** |
| | Date | 3, 35 | 4.77 | **0.007** |
| | Treatment:Date | 3, 35 | 1.24 | 0.310 |
| **Logistic Regression** | | | | |
| | | **DF** | **$\chi 2$** | **P** |
| **Survivorship*** | Treatment | 1 | 0 | 0.995 |
| | Species | 8 | 15.859 | **0.044** |
| | Treatment:Species | 8 | 15.037 | 0.058 |

Results from statistical analyses. Results from SAS software are indicated by an asterisk*. All other results come from R.

of the relative change in width indicated no significant difference depending on treatment, nor a treatment-by-species interaction. There was a significant species effect, with *Encelia californica* exhibiting the greatest relative growth in width and *Malosma laurina* experiencing the least relative change in width (Fig 3F).

Survivorship at the end of the study, analyzed by logistic regression, indicated that *Baccharis emoryi* and *Salvia mellifera* had higher survival than *Malosma laurina* and *Atriplex lentiformis* (Fig 3G), but that there was no significant effect of treatment and there was a marginally significant treatment-by-species interaction. There was a significant treatment-by-species interaction for shrub lifespan (Fig 3H). *Acmispon glaber* and *Atriplex lentiformis* lived longer when seeded. All other species either had longer lifespans when planted or not enough surviving individuals to compare.

## Cost comparison

Restoration efforts using container plants were more time-consuming than through seed. Restoring with self-grown plants required about 25% more hours of labor compared to seeding, with this difference largely due to the time associated with growing out plants in the

**Table 2. Summary of functional traits and results by treatment.**

| Species | Functional Traits | | | | Experimental Results | | | | | | |
|---|---|---|---|---|---|---|---|---|---|---|---|
| | Leaf Lifespan | Flowering Time | Root Depth | Post-fire Strategy | % Cover | Density | RGR Height | RGR Width | Days Alive | % Survivorship | Overall |
| **All Shrub Species Combined** | - | - | - | - | NS | **S** | **S** | NS | **P** | **P** | S&P |
| *Acmispon glaber* | Deciduous | Mar-Aug | Shallow | Obligate Seeder | S | **S** | **S** | S | **S** | **S** | S |
| *Artemisia californica* | Deciduous | Aug-Nov | Shallow | Facultative Seeder | NS | **S** | **S** | S | **P** | **P** | S&P |
| *Atriplex lentiformis* | Deciduous | July-Oct | Variable | NA | S | **S** | **NA** | P | **S** | **S** | S |
| *Baccharis emoryi* | Evergreen | May-Nov | Moderate/ Deep | Crown Sprouter | NS | **P** | **S** | P | **P** | **P** | S&P |
| *Elymus condensatus* | Evergreen | Jun-Aug | Shallow | Crown Sprouter | P | **P** | NA | NA | NA | NA | P |
| *Encelia californica* | Deciduous | Feb-Jun | Shallow | Facultative Seeder | S | **S** | **S** | NS | **P** | **P** | S&P |
| *Eriogonum fasciculatum* | Somewhat Deciduous | All Year | Shallow/ Moderate | Facultative Seeder | S | **S** | **S** | S | **P** | **P** | S&P |
| *Isocoma menziesii* | Evergreen | Jun-Nov | Deep | Rare Crown Sprouter | P | **P** | NA | NA | NA | NA | P |
| *Malosma laurina* | Evergreen | Jun-Jul | Deep | Facultative Seeder | P | **S** | **P** | P | **P** | **P** | S&P |
| *Peritoma arborea* | Evergreen | All Year | Variable | NA | NS | **S** | **S** | S | **P** | NS | S&P |
| *Rhus integrifolia* | Evergreen | Feb-May | Deep | Facultative Seeder | P | **S** | NA | NA | NA | NA | S |
| *Salvia mellifera* | Deciduous | Mar-Jun | Shallow | Facultative Seeder | S | **P** | **S** | P | **P** | **P** | S&P |

Summary of functional traits and results, with the treatment in which each species did best (S = Seeded, P = Planted, S&P = mixed results, NS = no difference, and NA = not available). The first row provides a summary of results for all species combined. Bold indicates differences that are statistically significant (p<0.05). Leaf lifespan describes the behavior of the species observed at or near the study site. Root depth is what is typical for the species, and categories are defined as follows: Shallow is <1.5m; Moderate is >1.5m and <3m; Deep is >3m; "Variable" indicates conflicting information in the literature or a large range of possible depths. For post-fire strategy, NA indicates that the species' behavior has not been well documented.

greenhouse (Table 3). While not as time-consuming as efforts using self-grown plants, restoring with nursery-grown plants was still more labor-intensive compared to seeding (Table 3). When comparing the amount of time it took to plant the native seedlings versus sowing native seed directly in the ground, the difference was marginal—only by about 11 hours. Although we saw differences in the number of hours dedicated to these tasks, it took the same amount of time to maintain the study area regardless of whether it was planted or seeded (Table 3).

More money was spent restoring with container plants than seed, with the bulk of the costs attributed to labor and supplies needed to grow plants (Table 3). In comparison to simply seeding, we spent 1.4 times more money to use plants that we grew in our own greenhouse and would have spent about 1.2 times more money to use plants purchased from a local nursery. When comparing the cost of perennial seed mixes between restoration techniques (self-grown plants versus seed), more money was spent on perennial seed mixes for the seeding method (Table 3). This is because the seeding method uses more seed to account for the possibility of loss through drift, predation, or inviable seed. Restoration is the most expensive using self-grown plants followed by nursery-grown plants, and lastly seeding (Table 3). Although seeding efforts required a larger volume of seed, the additional cost of supplies and labor associated with nursery-grown plants and self-grown plants resulted in these techniques being more expensive overall.

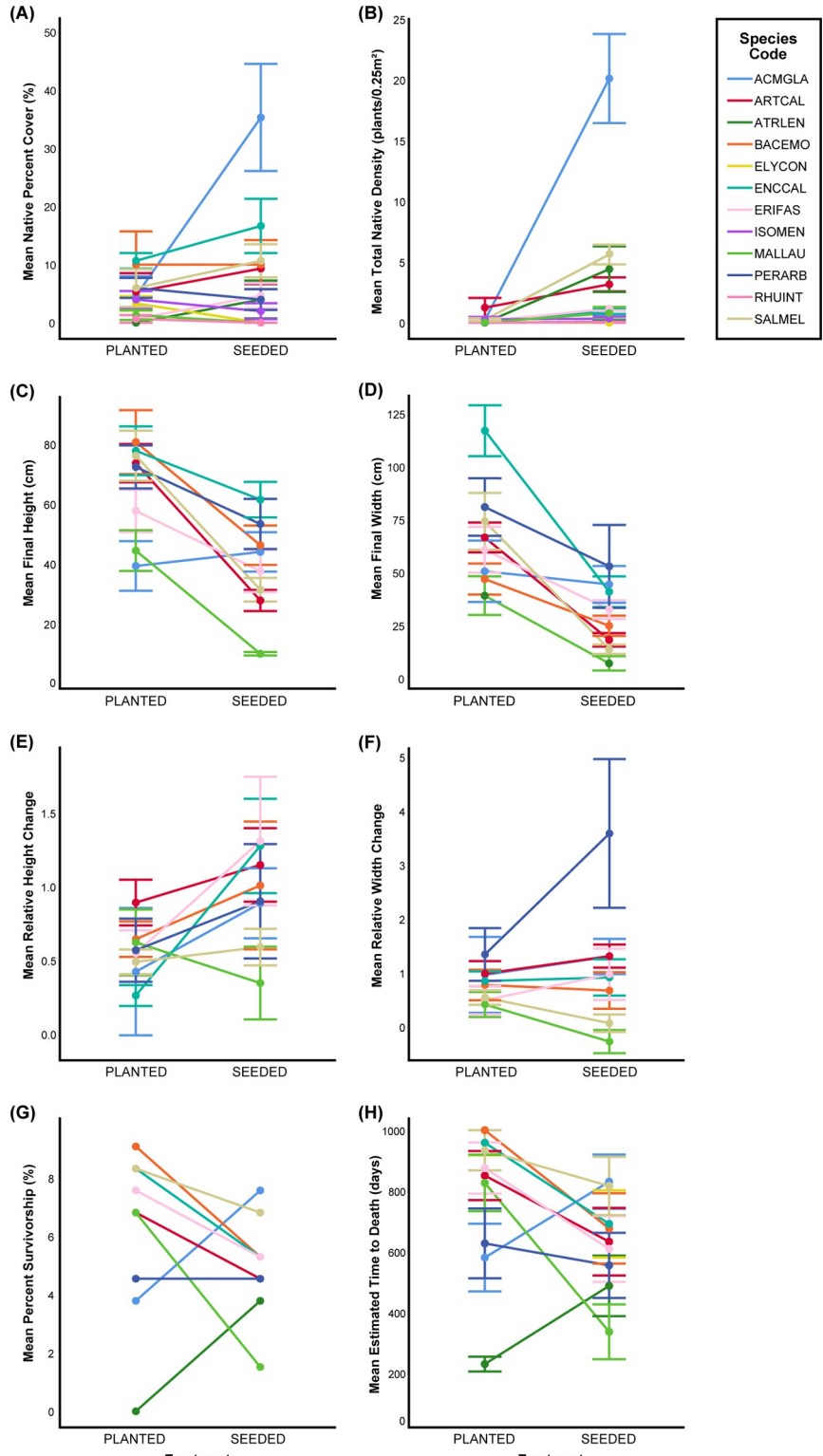

**Fig 3. Average values for each species in the two different treatments.** Dots indicating average values for each shrub mix species in each treatment with bars representing mean +/− 1 SE. Data shown includes A) native percent cover, B) native density, C) final height, D) final width, E) relative height change, F) relative width change, G) percent survivorship, H) estimated time to death. Species with very low sample sizes were excluded from these analyses.

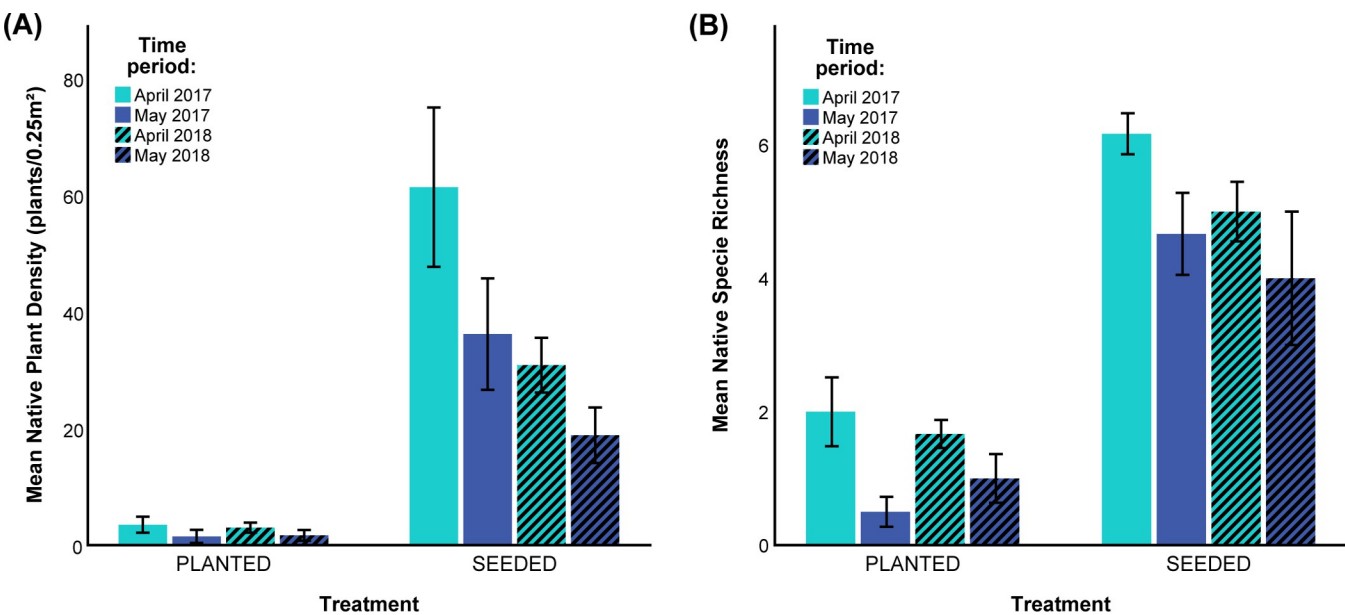

**Fig 4. Density and species richness over time.** A) Density and B) species richness values from four different time points during which these values were measured. Bars represent means +/− 1 SE.

## Discussion

Our finding that the success of restoration from seeds compared to container plants varied depending on both the species and the metric of success (i.e. native cover vs. density). This supports other studies that have emphasized varying results depending on those contextual variables [83,84]. While restoration from seeds resulted in greater density, relative growth, richness, and cost-effectiveness than restoration from container plants, container plants overall had greater survivorship than plants added to the site as seeds, consistent with findings from a meta-analysis of studies comparing restoration from seeds and plants [12].

Seeded plots outperformed planted plots in terms of native shrub density, but this trend was driven by one species, *Acmispon glaber*. Shrubs in seeded plots had a greater relative change in height but had lower final height than their planted counterparts even after two

**Table 3. Cost comparison of different restoration techniques.**

| Time Cost (hours) | | | | | | |
|---|---|---|---|---|---|---|
| Restoration Technique | Plant Grow-Out | Planting | Seed Mix Prep | Seeding | Site Maintenance | Total Hours |
| Seed | 0 | 0 | 72.25 | 110 | 590.25 | 772.5 |
| Self-Grown Plants | 239.25 | 121.5 | 11.5 | 0 | 590.25 | 962.5 |
| Nursery Plants | 0 | 121.5 | 0 | 0 | 590.25 | 711.8 |
| Monetary Cost ($) | | | | | | |
| Restoration Technique | Contracted Grow-Out—Perennial | Perennial Grow-Out Supplies | Perennial Seed Mix | Labor | Total Cost | |
| Seed | 0.00 | 0.00 | 94.92 | 14932.43 | 15027.4 | |
| Self-Grown Plants | 0.00 | 1259.47 | 2.41 | 19867.01 | 21128.9 | |
| Nursery Plants | 2156.45 | 0.00 | 0.00 | 15914.58 | 18071.0 | |

This table reports the total amount of hours and money spent on restoration using seed and self-grown plants during the first year of the study (2016–2017). It also estimates the cost of using nursery-grown plants.

growing seasons. This, as well as the greater mortality of seeded plants, could be due to our high seeding rates. Competition of many emerging plants packed closely together may have limited plant growth and lowered growth and survival in the seeded treatment [85]. Determining seeding rates that are high enough to fill open space without leading to decreased growth due to competitive interactions is one of the complexities of conducting restoration from seed [24]. The higher mortality of seeded shrubs may also be because they were younger and less developed and therefore may not have been able to tolerate environmental stress as well as their planted counterparts [86].

Native richness is frequently lower in restored systems compared to intact systems and increasing native richness is a goal of many restoration projects [87,88]. Not only was richness greater for plots containing native shrubs added to the site as seeds compared to container plants, but restoration from seed allowed for the addition of a diversity of annual species [89]. Despite greater overall richness with seeds, there were species-specific responses, perhaps due to physiological and life-history traits, that influenced performance when added to the site as seeds or as container plants.

The two species that established best from seed in our study, *Atriplex lentiformis* and *Acmispon glaber*, are both drought-deciduous at our site, and they are both known to colonize areas from seed following wildfires [90,91]. *A. glaber* had a greater density and percent cover than any other species in the seeded treatment, likely due to its unintentionally high seeding rate. This species drove the trends in seeded plot cover and density. Its role in the coastal sage scrub community as an important early-successional fire-follower [92] and as a species that grows quickly in recently cleared shrublands [93] may have also contributed to *A. glaber*'s success when seeded. Further studies could investigate this possibility.

The two deep-rooted, evergreen species with sclerophyllous leaves—*Malosma laurina* and *Rhus integrifolia*—did not establish well from seed. Despite being seeded at a moderate rate, no *R. integrifolia* and only three *M. laurina* plants sprouted and survived in the seeded treatment. This could be due to the species having endogenous seed dormancy, which may not have been broken by our one month of cold stratification, possibly resulting in a low number of germinants (S1 Table). *M. laurina* is also known to sprout later than other shrub species, so competition may have been a limiting factor in the seeded plots [71]. Restoration from plants rather than from seed seems to be more common for these two species, and few studies show successful restoration from seed [72]. Our results support the continued use of container plants when adding *M. laurina* and *R. integrifolia* to a restoration site, and suggest that practitioners establishing large, deep-rooted shrubs may have better results using container plants rather than seeds.

The failure of *Isocoma menziesii* to grow in the seeded treatment was more surprising since it is known to readily establish from seed [69]. *Elymus condensatus*, a very large-statured perennial grass that we included in the shrub plots, likewise did not emerge from seed in our seeded plots but was successful in the container plant treatment. This aligns with previous studies that have successful establishment from *E. condensatus* plants [94] and poor establishment from seeds [95]. For species that did not emerge in the seeded plots, it's possible that the temperature and water requirements for germination were not met in the field [96,97]. However, all of the species germinated well in the greenhouse, and we were able to grow out container plants from the same seed lots. The majority of the shrub species in our study are relatively shallow-rooted, facultative seeders that did not have a definitive overall preference for seeding or planting (Table 2). There were also no clear trends between leaf lifespan or phenology and success in either treatment [98].

As expected, seeding efforts required less of a time commitment. When we look at the breakdown of the total hours spent on each restoration technique, we found that the largest

time commitment for both methods was associated with tasks related to site maintenance. This result echoes the findings of other studies that commented on the high cost of labor associated with maintaining the experimental site after it has been seeded or planted [5,12,99,100]. Our sowing efforts, which involved hand sowing followed by raking and tamping, is one of the most labor-intensive and time-consuming seeding methods [5]. The combined time and effort associated with growing out seedlings and then planting them in the field was greater than the time it took to simply seed the plots.

When comparing the monetary cost of restoration between restoration techniques, planting was much more expensive than seeding. Restoration efforts using container plants will generally be more expensive due to the added cost of supplies and labor associated with growing and tending to the plants until it is time for planting at the restoration site [12,101]. However, if container plants are necessary for successful restoration then we found that nursery-grown plants may be a cheaper alternative to plants grown at your own facility since the labor costs associated with caring for the plants would be outsourced to nursery staff.

Our findings suggest that restoration practitioners can often achieve cost-effective, successful restoration using seeds, which aligns with the conclusions of others [31,102]. However, the use of container plants allows for greater control of community composition [85], in particular, for those species that do not readily recruit from seed. Using a combination of seeding most species and planting the species that do not recruit easily from seed may be optimal for restoring native richness to a community.

## Supporting information

**S1 Fig. Example block layout.** An example of an experimental block at the site. The block includes plots that accommodate all seed mixes, including two larger shrub mix plots, two forb plots, and four grass mix plots (two each for the two grassland species, *Stipa pulchra* and *Stipa lepida*, used in this study.) One half of the block was randomly assigned to the planting treatment, while the other half was seeded. Forb plots were seeded in both halves of the plots.
(TIF)

**S2 Fig. Density in all plot types.** Density of plants seeded and planted into the respective plot types in A) May 2017 and B) May 2018. (Density data was not collected in the grass plots [STI-LEP and STIPUL] in May 2018. Bars represent mean +/− 1 SE.
(TIF)

**S3 Fig. Percent cover values for all plants found at the site.** Mean percent cover of native plants that were on the shrub mix species list compared to volunteer plants that were not on the species list and established naturally. Volunteer plants are further broken down into natives and non-natives. Bars represent mean +/− 1 SE.
(TIF)

**S4 Fig. Number of tagged plants per species.** The number of tagged shrub mix individuals by species that were alive at the end of the study.
(TIF)

**S1 Table. Complete species list with associated seeding rates.** This table reports the percent pure live seed (PLS), seeding rate (seeds/m$^2$), and dormancy breaking procedure used in preparation for seeding or planting efforts. Information on other restoration projects in the area was used to determine seeding rates.
(DOCX)

**S2 Table. Description of restoration tasks.** Specific activities or materials associated with each generalized restoration task or purchase conducted in our study.
(DOCX)

## Acknowledgments

This study is located on the shared ancestral homeland of the Acjachemen and Tongva Peoples. Thanks to CEB undergraduate interns and Project Grow volunteers, who worked to implement the restoration and maintain the plots. Thanks to Anita Garg for helping collect data as part of her science fair project.

## Author Contributions

**Conceptualization:** Jennifer J. Long, Sarah Kimball.

**Data curation:** Katharina T. Schmidt, Priscilla Ta.

**Formal analysis:** Kylie D. F. McGuire, Sarah Kimball.

**Funding acquisition:** Matthew Yurko.

**Investigation:** Katharina T. Schmidt, Priscilla Ta.

**Methodology:** Priscilla Ta, Sarah Kimball.

**Project administration:** Priscilla Ta, Jennifer J. Long, Matthew Yurko, Sarah Kimball.

**Resources:** Matthew Yurko, Sarah Kimball.

**Supervision:** Katharina T. Schmidt, Priscilla Ta, Jennifer J. Long, Matthew Yurko, Sarah Kimball.

**Visualization:** Kylie D. F. McGuire, Katharina T. Schmidt.

**Writing – original draft:** Kylie D. F. McGuire, Katharina T. Schmidt, Priscilla Ta, Sarah Kimball.

**Writing – review & editing:** Kylie D. F. McGuire, Katharina T. Schmidt, Priscilla Ta, Sarah Kimball.

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
