## [Decision Letter · Decision Letter 0]

3 Feb 2021

PONE-D-20-35684

Is it best to add native plants to restoration projects as seeds or as seedlings?

PLOS ONE

Dear Dr. McGuire,

Thank you for submitting your manuscript to PLOS ONE. After careful consideration, we feel that it has merit but does not fully meet PLOS ONE’s publication criteria as it currently stands. Therefore, we invite you to submit a revised version of the manuscript that addresses the points raised during the review process.

Taking into account reviewers’ comments, some of the most relevant and shared concerns are:

* Consider reducing the length of paper (ca. 25%), avoid repetitions, and make most parts more simple and straightforward, particularly the introduction and results sections.

* Reconsider the inclusion of some variables and treatments (e.g., chlorophyll content and the forbs seeded treatment) or discuss in the reply letter the reasons for keeping them in your analyses.

* If possible, add some missing information that may improve the discussion of your results, e.g., include the difference in total hours for the plant versus seed method in the Total Cost estimate; seeding rate of each species (# seeds not mass) and the viability of each species.

We look forward to receiving your revised manuscript.

Kind regards,

Cristina Armas

Academic Editor

PLOS ONE

Additional Editor Comments:

Academic Editor:

Please, use SI units throughout the text and figures (mm for rainfall, m or cm for length/diameter and so on)

L. 123 and Fig. 2 and other parts of main text - Use SI units for rainfall, e.g., 317 mm.

L. 175 Diameter in cm or m

Minor point. Consider including a couple of tables in the main text (or appendices) with results from the statistical analyses. That will make the result section more straightforward without so many parentheses including the statistical results. The authors may also replace some of these data in the main text with mean +/- error values.

"This study was supported by the Center for Environmental Biology (CEB) at the University of

484 California, Irvine (UCI) and by Project Grow, a non-profit project of the California Coastal

485 Commission and the Tides Center."

4. We note that Figure 1 in your submission contains satellite images which may be copyrighted. All PLOS content is published under the Creative Commons Attribution License (CC BY 4.0), which means that the manuscript, images, and Supporting Information files will be freely available online, and any third party is permitted to access, download, copy, distribute, and use these materials in any way, even commercially, with proper attribution. For these reasons, we cannot publish previously copyrighted maps or satellite images created using proprietary data, such as Google software (Google Maps, Street View, and Earth). For more information, see our copyright guidelines: http://journals.plos.org/plosone/s/licenses-and-copyright.

(1) You may seek permission from the original copyright holder of Figure 1 to publish the content specifically under the CC BY 4.0 license. 

Reviewers' comments:

Reviewer's Responses to Questions

**Comments to the Author**

1. Is the manuscript technically sound, and do the data support the conclusions?

Reviewer #1: Partly

Reviewer #2: Yes

2. Has the statistical analysis been performed appropriately and rigorously? 

Reviewer #1: Yes

Reviewer #2: Yes

3. Have the authors made all data underlying the findings in their manuscript fully available?

Reviewer #1: Yes

Reviewer #2: No

4. Is the manuscript presented in an intelligible fashion and written in standard English?

Reviewer #1: Yes

Reviewer #2: Yes

5. Review Comments to the Author

Reviewer #1: TITLE

The title is too generic for a study developed only in one site. It is necessary to clarify which types of native plants have been used (trees, shrub, grass…?) and which types of ecosystems have been restored (forests, savannas, grasslands…?).

ABSTRACT

Line 24 – It is necessary to clarify the term “species identity”. I do not think that the influence of the weather in the outcomes has been analyzed in the study.

Line 24 – The authors cited that that measured percent cover, density, diversity (probably they meant species richness), height… However, in the abstract they only showed and discussed the density and survival.

Line 32 – According to authors, the responses varied by species. The evaluation of species performances has not been cited as one of the objectives of the study. As a matter of fact, the objectives of the study are not clear in the abstract.

Line 34 – The conclusions and the objectives of the study are not properly linked. Moreover, the measurements made do not support this conclusion.

INTRODUCTION

Lines 47 – It is essential to explain which type of ecosystem the study aim to restore.

Lines 49 – 51 – This sentence is unnecessary, as the experiment did not study weed management.

Lines 55 – 56 – It is necessary to add a reference.

Lines 57 – 60 – It is necessary to add references. For example: Doust, S.J., Erskine, P.D., Lamb, D., 2006. Direct seeding to restore rainforest species: microsite effects on the early establishment and growth of rainforest tree seedlings on degraded land in the wet tropics of Australia. Forest Ecology and Management 234, 333-343; Doust, S.J., Erskine, P.D., Lamb, D., 2008. Restoring rainforest species by direct seeding: Tree seedling establishment and growth performance on degraded land in the wet tropics of Australia. Forest Ecology and Management 256, 1178-1188; Meli, P., Isernhagen, I., Brancalion, P.H.S., Isernhagen, E.C.C., Behling, M., Rodrigues, R.R., 2018. Optimizing seeding density of fast-growing native trees for restoring the Brazilian Atlantic Forest. Restoration Ecology 26, 212-219; Souza, D.C., Engel, V.L., 2018. Direct seeding reduces costs, but it is not promising for restoring tropical seasonal forests. Ecological Engineering 116, 35-44.

Lines 60 – 64 – These sentences are unnecessary, as the experiment did not study seed dormancy break.

Lines 66 – 68 – These sentences need references.

Line 76 – 77 – Add reference.

Lines 86 – 88 – Add reference.

Lines 89 – 93 – These sentences are unnecessary.

Lines 105 – 109 – This sentence is not clear. As a matter of fact, I think that the objectives section is too long, unclear, and a little disconnected to the methods. I suggest rewriting the entire section.

MATERIALS AND METHODS

Lines 122 – 123 – Add reference.

Lines 138 – 144 – This sentence is not clear.

Lines 145 – 150 – If different species was used in different years, the manuscript should not discuss the effects of weather on the outcomes.

Lines 179-180 – Only two individuals? That is a very small number of individuals.

Line 182 – Forbs were added just as seeds? So, there are no reasons to use them considering the main objective of the experiment.

Line 193 – If the data have not been included in the analyses, there are no reasons of cite it.

Line 198 – In the abstract and introduction sections the authors used the term “diversity” to refer to “species richness”. It is necessary to correct it.

Line 200 – Considering the objectives of the study, to evaluate the chlorophyll content is not necessary. This measurement was not important to discussion and conclusions.

Line 223 – I suggest eliminating this measurement.

RESULTS

Line 272 – It is not clear how seeded/planted individuals were differentiated from natural regeneration.

Line 282 – If the focus was on the shrub species, why to use forbs and grasses?

Line 291 – If the data is not statistical significant, it is not necessary to cite that they were higher or lower.

Line 295 – Explain “identity”.

Lines 338 – 340 – This is obvious. It is not necessary to present these results. I suggest removing this sentence and the next one.

DISCUSSION

Lines 403 – 405 – Your results do not support this affirmation.

Lines 409 – 411 – The measurement of chlorophyll content was not necessary.

Lines 414 – 416 – This was expected.

Lines 416 – 417 – Your results do not support this affirmation.

Line 422 – Diversity or species richness?

Line 425 – It is necessary to better explain the species-specific responses in the objectives, materials and methods, and results section.

Lines 453 – 455 – Add references. It is necessary to better explain this sentence.

Lines 457 – 458 – Your objectives and results are not clear about the evaluation of functional traits in your study. It is necessary to better explain it.

GENERAL OVERVIEW

The comparison of restoration efforts by seeds and seedlings is very important for restoration practitioners and the manuscript could improve the discussion. However, the manuscript is not clear and the objectives, results, discussion and conclusions are not entirely connected. To rewrite the objective section is essential, and also to better explains the results according to the objectives. There are several unnecessary sentences, while important analyses are not wee-explained. The manuscript only will be appropriate for publication after substantial changes.

Reviewer #2: This paper reports results of a restoration experiment conducted in a coastal sage scrub plant community in Southern California, USA. The primary goal of the study was to determine whether seeding or planting was more effective on establishment and other measures of restoration success. The study also compared the treatments among 12 species and reported costs for each treatment. This study reports new experimental findings. The experiment and analysis appear to be sound. My main suggestion for improvement is to reduce the length of the paper by about 25-30%. The study was fairly simple and straightforward, however, the paper reads long and is somewhat repetitive. Also, parts of the writing and language used could be tightened up and made more professional. The difference in hours required for the methods should be added into the Total Cost of each method. This will make the planting costs much more expensive, so this will need to be revised in the discussion. Specific comments by line number follow.

78 correct typo “plants added by direct seeded”

90-91 “that you hope to” and “you’re adding” are too informal and should be avoided

186 “making sure to allow each plant enough space to avoid competition” – packing 12 shrubs into a 15 square meter area does not seem like enough space to avoid competition for the mature plants. Do you mean for the seedlings only? Why would you plant them so close? Were you expecting some to die?

223 It is not clear why chlorophyll content was measured, and measured only once. Chlorophyll content would vary among species (which does not really relate to this study) or between different resource environments (such as a wet year and a dry year – which would require more data). The data are also not really discussed since there were no treatment effects. I suggest removing from the paper since it does not add anything to the interpretation.

238 – 245 The description of the treatments is somewhat confusing throughout the paper. There were shrubs that were seeded, shrubs that were planted, grasses that were seeded, grasses that were planted, and forbs that were seeded. It is really confusing to say shrubs planted in combination with seeded forbs as this implies they are put in the same plot. I found this really confusing throughout the methods, results, and discussion. Consider revising.

247 Data analysis – there are t statistics reported in the results but I cannot figure out which tests they go with? Were there t tests? If so, they should be described in this section. For models with species were the treatment x species interaction terms included? Say that in this section.

270, 273 The mention of two years of non-native removal here was very confusing. There was no discussion of non-native removal before this point. Include this in the methods or remove the reference here.

281 Here it says the focus is on the native shrubs, then why are there so many other grass and forb treatments? Would it be more straightforward to remove the grasses and forbs from this paper? Or is there a reason for including them? It is not really very clear to me.

292, 339 Here are some t results. Please indicate which of the tests from the methods these go with. Based on the described analyses it seems like F statistics should be reported instead. Refer to figures in order in text (Fig 2 should come before Fig 3, etc.)

298 revise this statement

325 “nor” is not the correct word here, revise

376 revise “not a huge difference” to be more scientific; What is the purpose of reporting the number of hours for different people? This was never made clear or added into the cost estimates.

Table 2 Was the difference in total hours for the plant versus seed method also included in the Total Cost estimate in the second half of the table. Labor is one of the most expensive aspects of restoration and these 190 hours would equate to $4000 plus that should be added to the planting costs. This needs to be revised here and in the discussion.

382 The descriptions here of the methods are confusing. Why are you evaluating methods that you did not use? This could be revised to be more clear or reduced for simplicity.

390 Why are we talking about planted versus seeded forb treatments at all? They were separate plots that were both seeded, one just happened to be closer to the planted shrubs. This is really confusing.

450 Throughout the paper I felt that I needed to know the seeding rate of each species (# seeds not mass) and the viability of each species. How did you determine the seeding rate for each species? Please add this information to S1 Table. It may be that the Elymus seeds had low viability and you can rule that out if you did a seed viability trial for each species. If you didn’t do this, it is OK but it limits your ability to interpret the data.

466 If you add the dollar amount of the time you will see that outplanting was much more costly.

Fig 1 add scale bar to plot design, make larger map smaller if possible

S1 Table – Please put seeding rate in # seeds/sq m for easier comparison with other studies; include viability for each species and how the seeding rate for each species was determined.

6. PLOS authors have the option to publish the peer review history of their article (what does this mean?). If published, this will include your full peer review and any attached files.

Reviewer #1: No

Reviewer #2: No

---

## [Author Response · Author response to Decision Letter 0]

19 May 2021

Responses to the Academic Editor:

Please, use SI units throughout the text and figures (mm for rainfall, m or cm for length/diameter and so on)

L. 123 and Fig. 2 and other parts of main text - Use SI units for rainfall, e.g., 317 mm.

L. 175 Diameter in cm or m

Response: Thank you for pointing this out. The rainfall units have been changed to mm, and the pot diameter units have been changed to cm.

Minor point. Consider including a couple of tables in the main text (or appendices) with results from the statistical analyses. That will make the result section more straightforward without so many parentheses including the statistical results. The authors may also replace some of these data in the main text with mean +/- error values.

Response: We have added a table with results from our statistical analyses to the main text of the paper (Table 1) to address this issue.

Response: Thank you. We have reviewed our submission to ensure that it adheres by these guidelines.

"This study was supported by the Center for Environmental Biology (CEB) at the University of

484 California, Irvine (UCI) and by Project Grow, a non-profit project of the California Coastal

485 Commission and the Tides Center."

Response: The statement has been removed from the Acknowledgments of the manuscript. We request that the financial statement read “This study was supported by the Center for Environmental Biology (CEB) at the University of California, Irvine (UCI) and by Project Grow, a non-profit project of the California Coastal Commission and the Tides Center.”

Response: Our data have been privately uploaded to the Dryad data repository. The temporary URL is https://datadryad.org/stash/share/9P51_ZTnr9GFFgltNXyfbruPWtmUfijsYDKJvHA67lU. 

The DOI (not yet live) is10.5061/dryad.95x69p8jm.

4. We note that Figure 1 in your submission contains satellite images which may be copyrighted. All PLOS content is published under the Creative Commons Attribution License (CC BY 4.0), which means that the manuscript, images, and Supporting Information files will be freely available online, and any third party is permitted to access, download, copy, distribute, and use these materials in any way, even commercially, with proper attribution. For these reasons, we cannot publish previously copyrighted maps or satellite images created using proprietary data, such as Google software (Google Maps, Street View, and Earth). For more information, see our copyright guidelines: http://journals.plos.org/plosone/s/licenses-and-copyright.

Response: We attempted to contact the copyright holders but received no response. Our team has determined that this figure added very little to the manuscript, so we have chosen to omit it from our revised manuscript altogether.

 

Responses to Reviewer 1:

Reviewer #1: TITLE

The title is too generic for a study developed only in one site. It is necessary to clarify which types of native plants have been used (trees, shrub, grass…?) and which types of ecosystems have been restored (forests, savannas, grasslands…?).

Response: We have edited the title as recommended to more specifically reference the type of community restored. It now reads,

“Is it best to add native shrubs to coastal sage scrub restoration projects as seeds or as seedlings?”

ABSTRACT

Line 24 – It is necessary to clarify the term “species identity”. I do not think that the influence of the weather in the outcomes has been analyzed in the study.

Response: We have edited the abstract to delete the mention of weather as recommended. The term “species” is used in place of “species identity.”

Line 24 – The authors cited that that measured percent cover, density, diversity (probably they meant species richness), height… However, in the abstract they only showed and discussed the density and survival.

Response: We have edited the Abstract to reflect all of the variables measured in the study. The sentence now says,

 “We measured percent cover, density, species richness, size, survival, and costs.”

Line 32 – According to authors, the responses varied by species. The evaluation of species performances has not been cited as one of the objectives of the study. As a matter of fact, the objectives of the study are not clear in the abstract.

Response: We have edited the Abstract to clarify the study objectives. It now reads, 

“Ecological restoration frequently involves the addition of native plants, but the effectiveness (in terms of plant growth, plant survival, and cost) of using seeds versus container plants has not been studied in many plant communities. It is also not known if plant success would vary by species or based on functional traits such as leaf lifespan. To answer these questions, we added several shrub species to a coastal sage scrub restoration site as seeds or as seedlings in a randomized block design.”

Line 34 – The conclusions and the objectives of the study are not properly linked. Moreover, the measurements made do not support this conclusion.

Response: We’ve edited the objectives as recommended to ensure that the conclusions are clearly linked. We’ve also clarified how our measurements supported these conclusions. 

INTRODUCTION

Lines 47 – It is essential to explain which type of ecosystem the study aim to restore.

Response: We have edited this sentence to specify that our study is on coastal sage scrub.

Lines 49 – 51 – This sentence is unnecessary, as the experiment did not study weed management.

Response: We have deleted this sentence as recommended. 

Lines 55 – 56 – It is necessary to add a reference.

Response: Upon further evaluation, we determined that this sentence was confusing, and actually contradicted some of the papers (such as Greet 2020). We chose to remove the sentence.

Lines 57 – 60 – It is necessary to add references. For example: Doust, S.J., Erskine, P.D., Lamb, D., 2006. Direct seeding to restore rainforest species: microsite effects on the early establishment and growth of rainforest tree seedlings on degraded land in the wet tropics of Australia. Forest Ecology and Management 234, 333-343; Doust, S.J., Erskine, P.D., Lamb, D., 2008. Restoring rainforest species by direct seeding: Tree seedling establishment and growth performance on degraded land in the wet tropics of Australia. Forest Ecology and Management 256, 1178-1188; Meli, P., Isernhagen, I., Brancalion, P.H.S., Isernhagen, E.C.C., Behling, M., Rodrigues, R.R., 2018. Optimizing seeding density of fast-growing native trees for restoring the Brazilian Atlantic Forest. Restoration Ecology 26, 212-219; Souza, D.C., Engel, V.L., 2018. Direct seeding reduces costs, but it is not promising for restoring tropical seasonal forests. Ecological Engineering 116, 35-44.

Response: The following references have been added:

Doust SJ, Erskine PD, Lamb D. Direct seeding to restore rainforest species: Microsite effects on the early establishment and growth of rainforest tree seedlings on degraded land in the wet tropics of Australia. For Ecol Manag. 2006;234: 333–343. doi:10.1016/j.foreco.2006.07.014

Applestein C, Bakker JD, Delvin EG, Hamman ST. Evaluating Seeding Methods and Rates for Prairie Restoration. Nat Areas J. 2018;38: 347–355. doi:10.3375/043.038.0504

Souza DC de, Engel VL. Direct seeding reduces costs, but it is not promising for restoring tropical seasonal forests. Ecol Eng. 2018;116: 35–44. doi:10.1016/j.ecoleng.2018.02.019

Lines 60 – 64 – These sentences are unnecessary, as the experiment did not study seed dormancy break.

Response: We have deleted these sentences as recommended. 

Lines 66 – 68 – These sentences need references.

Response: Appropriate references have been added.

Wainwright CE, Cleland EE. Exotic species display greater germination plasticity and higher germination rates than native species across multiple cues. Biol Invasions. 2013;15: 2253–2264. doi:10.1007/s10530-013-0449-4

Shaw N, Barak RS, Campbell RE, Kirmer A, Pedrini S, Dixon K, et al. Seed use in the field: delivering seeds for restoration success. Restor Ecol. 2020;28. doi:10.1111/rec.13210

Erickson VJ, Halford A. Seed planning, sourcing, and procurement. Restor Ecol. 2020;28. doi:10.1111/rec.13199

Line 76 – 77 – Add reference.

Response: Appropriate references have been added.

Mortlock BW. Local seed for revegetation. Where will all that seed come from? Ecol Manag Restor. 2000;1: 93–101. doi:10.1046/j.1442-8903.2000.00029.x

Lines 86 – 88 – Add reference.

Response: Appropriate references have been added.

Filho PL, Leles PS dos S, Abreu AHM de, Fonseca AC da, Silva EV da. Seedling production of Ceiba speciosa in different volume of tubes using biosolids as substrate . Ciênc Florest. 2019;29: 27–39. doi:10.5902/1980509819340

Lines 89 – 93 – These sentences are unnecessary.

Response: We have deleted these sentences as recommended. 

Lines 105 – 109 – This sentence is not clear. As a matter of fact, I think that the objectives section is too long, unclear, and a little disconnected to the methods. I suggest rewriting the entire section.

Response: This sentence has been removed. The objectives paragraph has been shortened and edited to improve clarity.

In this study, we aim to understand the most effective way to add native shrub species to a restoration site - as seed or as container plants. Specifically, we hope to answer the following questions comparing plots in which native shrubs were added as seeds or as container plants: (1) Which is the most successful restoration technique for: a) increasing the percent cover, richness, and density of native plants?; and b) increasing survivorship and growth of native shrubs?; (2) Which restoration technique costs the least money or hours of labor? (3) Are the results consistent or do they vary by species and by functional traits? We hypothesized that container plants would be best for increasing cover, while seeds would be best for increasing richness and that the plants grown from seed might grow faster and have higher survivorship due to improved root health. We also hypothesized that restoration from seed would be less expensive and time-consuming than restoration from container plants. Finally, we did not expect the plants’ success to vary significantly by species or functional traits

MATERIALS AND METHODS

Lines 122 – 123 – Add reference.

Response: Appropriate references have been added.

Cleland EE, Funk JL, Allen EB. TWENTY-TWO. Coastal Sage Scrub. Ecosystems of California. University of California Press; 2016. pp. 429–448. Available: https://www.degruyter.com/document/doi/10.1525/9780520962170-026/html

Lines 138 – 144 – This sentence is not clear.

Response: This section has been reworded to improve clarity.

Each block contained two shrub plots (3 m x 5 m), four grass plots (1 m x 5 m), and two forb plots (1 m x 6 m). All twelve shrub species were added to the two shrub plots. Of the four grass plots, Stipa pulchra was added to two plots. Stipa lepida was added to the other two. Grass and shrub plots were randomly assigned to a seeded or planted treatment. Because annual forbs are rarely, if ever, added to restoration sites as plants, we decided against studying the relative success of planted and seeded forbs. Instead, all forbs were added to the site as seeds, with the intent of increasing native plant diversity and creating a healthier restored ecosystem than could have been achieved by using shrubs and grasses alone (S1 Fig). 

Lines 145 – 150 – If different species was used in different years, the manuscript should not discuss the effects of weather on the outcomes.

Response: All references to weather have been removed.

Lines 179-180 – Only two individuals? That is a very small number of individuals.

Response: This sentence has been edited to clarify the total number of individuals of each species used.

In late January of 2017, two individuals of each of the twelve shrub species were added to each of the seven planted treatment shrub plots, for a total of 14 planted individuals per shrub species (S1 Table).

Line 182 – Forbs were added just as seeds? So, there are no reasons to use them considering the main objective of the experiment.

Response: We have clarified the reason for using forbs in the study in the Experimental Design section.

Because annual forbs are rarely, if ever, added to restoration sites as plants, we decided against studying the relative success of planted and seeded forbs. Instead, all forbs were added to the site as seeds, with the intent of increasing native plant diversity and creating a healthier restored ecosystem than could have been achieved by using shrubs and grasses alone (S1 Fig). 

Line 193 – If the data have not been included in the analyses, there are no reasons of cite it.

Response: References to the year 2 blocks, which were not analyzed, have been removed from the paper.

Line 198 – In the abstract and introduction sections the authors used the term “diversity” to refer to “species richness”. It is necessary to correct it.

Response: Thank you for telling us this. All instances of “diversity” have been replaced with “species richness.”

Line 200 – Considering the objectives of the study, to evaluate the chlorophyll content is not necessary. This measurement was not important to discussion and conclusions.

Response: We agree with this and have removed the evaluation of chlorophyll content from the manuscript.

Line 223 – I suggest eliminating this measurement.

Response: We agree with this and have removed the evaluation of chlorophyll content from the manuscript.

RESULTS

Line 272 – It is not clear how seeded/planted individuals were differentiated from natural regeneration.

Response: Due to the invaded nature of our site, we expected that there would be little natural regeneration of native shrubs. This has been clarified in the Data Collection section:

“For the planted treatment plots, we flagged and monitored the individuals we planted. For the seeded treatment plots, we selected the healthiest looking individuals to monitor over time. Because of the invaded nature of the site, we assumed that shrub germinants were from the seeds that we added and not from natural regeneration.”

Line 282 – If the focus was on the shrub species, why to use forbs and grasses?

We have elaborated on this in the methods section.

Because annual forbs are rarely, if ever, added to restoration sites as plants, we decided against studying the relative success of planted and seeded forbs. Instead, all forbs were added to the site as seeds, with the intent of increasing native plant diversity and creating a healthier restored ecosystem than could have been achieved by using shrubs and grasses alone (S1 Fig). 

Line 291 – If the data is not statistical significant, it is not necessary to cite that they were higher or lower.

We have removed this from the sentence.

The total percent cover of seeded shrubs was marginally greater than the total percent cover of planted shrubs (Table 1 and Fig 2A).

Line 295 – Explain “identity”.

We agree that this was confusing, and have replaced all instances of the phrase “species identity” with simply “species.”

When the species of the native shrub was included as a fixed factor in the analysis of percent cover data in seeded and planted shrub plots, there was a significant interaction between species and treatment (Fig 3A).

Lines 338 – 340 – This is obvious. It is not necessary to present these results. I suggest removing this sentence and the next one.

Response: These sentences have been removed as recommended.

DISCUSSION

Lines 403 – 405 – Your results do not support this affirmation.

Response: We have tried to explain our findings more clearly throughout the paper and have edited the sentence to make our meaning clearer. We believe that the results do support this statement:

“Our finding that the success of restoration from seeds compared to container plants varied depending on both the species and the metric of success (i.e. native cover vs. density). This supports other studies that have emphasized varying results depending on those contextual variables [81,82].”

Lines 409 – 411 – The measurement of chlorophyll content was not necessary.

Response: The measurement of chlorophyll has been removed from the manuscript.

Lines 414 – 416 – This was expected.

Response: We believe that this result was not necessarily expected, since the significant difference in final size was observed after 2 years of growth at the site. The plants were started from seed at approximately the same time, so the difference in size was somewhat surprising to us. 

Lines 416 – 417 – Your results do not support this affirmation.

Response: We have qualified this statement to make it clear that we are suggesting a possible cause, rather than making an affirmation. The revised paragraph reads:

“Although seeded plots outperformed planted plots in terms of native shrub density, and shrubs in those plots had greater relative change in height, seeded shrubs had lower final height than their planted counterparts, even after two growing seasons. This, as well as the greater mortality of seeded plants, could be due to high seeding rates. Competition of many emerging plants packed closely together may have limited plant growth and lowered growth and survival in the seeded treatment [82]. Determining seeding rates that are high enough to fill open space without leading to decreased growth due to competitive interactions is one of the complexities of conducting restoration from seed [24].”

Line 422 – Diversity or species richness?

Response: Thank you for pointing this out. All instances of “diversity” have been replaced with “species richness.”

Line 425 – It is necessary to better explain the species-specific responses in the objectives, materials and methods, and results section.

Response: We have revised the manuscript to try to make our analysis of the species-specific responses more clear. 

The Objectives section now states that one of our goals is to determine “Are the results consistent or do they vary by species and by functional traits?”

The Materials and Methods section now says, “Each block contained two shrub plots (3 m x 5 m), four grass plots (1 m x 5 m), and two forb plots (1 m x 6 m). In order to determine if the response to seeding versus planting depended on the species of shrub used, we included twelve native shrub species in each shrub plot.”

We have also made sure to elaborate on which species performed better in which treatment for each contextual variable in the Results section, when significant.

Lines 453 – 455 – Add references. It is necessary to better explain this sentence.

Response: A reference has been added

Hardegree SP, Cho J, Schneider JM. Weather variability, ecological processes, and optimization of soil micro-environment for rangeland restoration. In: Monaco TA, Sheley RL, editors. Invasive plant ecology and management: linking processes to practice. Wallingford: CABI; 2012. pp. 107–121. doi:10.1079/9781845938116.0107

We have qualified this sentence (replacing “likely” with “possibly”) and have clarified that we have reason to believe that field conditions were a limiting factor, since the same seeds were able to grow well in the greenhouse.

“For species that did not emerge in the seeded plots, it’s possible that the temperature and water requirements for germination were not met in the field [citation]. However, all of the species germinated well in the greenhouse, and we were able to grow out container plants from the same seed lots.”

Lines 457 – 458 – Your objectives and results are not clear about the evaluation of functional traits in your study. It is necessary to better explain it.

Response: We have revised the Objectives section to explicitly state our intent to evaluate functional traits. It now reads, “(3) Are the results consistent or do they vary by species and by functional traits?”

We modified Table 1 to include all of the functional traits that we evaluate. We have chosen to organize the paper with the results from our measurements of cover, density, richness, size, and survivorship in the Results sections. We explain if individual species had greater success in the planted or seeded treatment, and then evaluate how their functional traits may relate to these results in the Discussion section.

 

Responses to Reviewer 2:

78 correct typo “plants added by direct seeded”

Response: Thank you for letting us know. This typo has been fixed.

90-91 “that you hope to” and “you’re adding” are too informal and should be avoided

Response: This sentence was deemed unnecessary and has been removed altogether.

186 “making sure to allow each plant enough space to avoid competition” – packing 12 shrubs into a 15 square meter area does not seem like enough space to avoid competition for the mature plants. Do you mean for the seedlings only? Why would you plant them so close? Were you expecting some to die?

Response: To clarify, there were actually 24 seedlings (2 individuals of each of the 12 species) in each planted plot - a density of 1.6 shrubs/m2. Our planting design was based on the knowledge of local restoration practitioners and from our own experiences with coastal sage scrub restoration projects. We did expect some shrubs to die, but even so, this is a fairly typical planting density for coastal sage scrub restoration.

223 It is not clear why chlorophyll content was measured, and measured only once. Chlorophyll content would vary among species (which does not really relate to this study) or between different resource environments (such as a wet year and a dry year – which would require more data). The data are also not really discussed since there were no treatment effects. I suggest removing from the paper since it does not add anything to the interpretation.

Response: We agree that this variable is not necessary and have removed all mentions of chlorophyll content from the manuscript.

238 – 245 The description of the treatments is somewhat confusing throughout the paper. There were shrubs that were seeded, shrubs that were planted, grasses that were seeded, grasses that were planted, and forbs that were seeded. It is really confusing to say shrubs planted in combination with seeded forbs as this implies they are put in the same plot. I found this really confusing throughout the methods, results, and discussion. Consider revising.

Response: We have revised the description in the Experimental Design section to better explain the treatments.

“Each block contained two shrub plots (3 m x 5 m), four grass plots (1 m x 5 m), and two forb plots (1 m x 6 m). All twelve shrub species were added to the two shrub plots. Of the four grass plots, Stipa pulchra was added to two plots. Stipa lepida was added to the other two. Grass and shrub plots were randomly assigned to a seeded or planted treatment. Because annual forbs are rarely, if ever, added to restoration sites as plants, we decided against studying the relative success of planted and seeded forbs. Instead, all forbs were added to the site as seeds, with the intent of increasing native plant diversity and creating a healthier restored ecosystem than could have been achieved by using shrubs and grasses alone. (S1 Fig).”

The sentence “Therefore the planting technique we actually implemented in our study was planting in combination with seeded forbs” has been removed, and the cost analysis section has been revised. This section now says:

 We also compared the total amount of money spent on materials for seeding or planting efforts during year one of our study to determine which technique was the least expensive. The specific materials purchased are stated in S2 Table. Since forb species were always sown directly on-site to mimic local restoration methods, we did not include the cost of restoring forbs in our analysis because there would be no difference. Labor is a major expense associated with any restoration effort so we also included the monetary cost of labor in our evaluation of the cheapest restoration technique [6,37–39]. We calculated the cost of labor assuming an hourly wage of $19.33 and using the total amount of time spent on preparations, seeding, planting, and site maintenance in our study [40]. While we did not use nursery-grown container plants in our study, we included this cost in our evaluation because nursery-grown container plants are often purchased and used in local restoration efforts. In our analysis, we compared the cost of restoration using seed, self-grown plants, and nursery-grown plants to determine which method costs the least amount of money.

247 Data analysis – there are t statistics reported in the results but I cannot figure out which tests they go with? Were there t tests? If so, they should be described in this section. For models with species were the treatment x species interaction terms included? Say that in this section.

Response: Thank you for pointing this out. This was an oversight on our end, and the Results section has been edited to include F values rather than t values.

270, 273 The mention of two years of non-native removal here was very confusing. There was no discussion of non-native removal before this point. Include this in the methods or remove the reference here.

Response: We have added a “Maintenance” section to the methods, which explains this:

“All blocks at the site were hand-weeded throughout the two years of the experiment to remove any non-natives that germinated at the site, with maintenance events occurring approximately twice a month during the winter and spring months.”

281 Here it says the focus is on the native shrubs, then why are there so many other grass and forb treatments? Would it be more straightforward to remove the grasses and forbs from this paper? Or is there a reason for including them? It is not really very clear to me.

Response: We have tried to better explain why shrubs, grasses, and forbs were all included in this project. Although we mainly studied the shrub plots, we believe that it was valuable to include other plants in the restoration experiment since the end result was a more diverse restored system.

The Experimental Design section now includes this clarification about the use of forbs:

Because annual forbs are rarely, if ever, added to restoration sites as plants, we decided against studying the relative success of planted and seeded forbs. Instead, all forbs were added to the site as seeds, with the intent of increasing native plant diversity and creating a healthier restored ecosystem than could have been achieved by using shrubs and grasses alone. (S1 Fig).

The Data Collection section now includes this clarifying section about the use of grasses:

Although our initial intent was to compare seeded and planted treatments for both shrubs and grasses, we observed very low germination and survival of all grasses. Because of this, we focused our data collection efforts on the shrub plots (S2 Fig).

292, 339 Here are some t results. Please indicate which of the tests from the methods these go with. Based on the described analyses it seems like F statistics should be reported instead. Refer to figures in order in text (Fig 2 should come before Fig 3, etc.)

Response: Thank you for pointing this out. This was an oversight on our end, and the Results section has been edited to include F values rather than t values. The order of figures in the citation has been corrected.

298 revise this statement

This sentence has been revise to simply read:

“When the species of the native shrub was included as a fixed factor in the analysis of percent cover data in seeded and planted shrub plots, there was a significant interaction between species and treatment (Fig 3A).”

325 “nor” is not the correct word here, revise

Response: This sentence has been edited to replace “nor” with “and.”

The density of plants decreased through time such that the effect of time of measurement was also significant and there was a significant treatment-by-year interaction.

376 revise “not a huge difference” to be more scientific; What is the purpose of reporting the number of hours for different people? This was never made clear or added into the cost estimates.

Response: This sentence has been removed. While we were originally interested in determining which technique required the most labor and how the labor was divided between the different work groups, we decided to remove this aspect from the manuscript because we realized our results would be confounded by factors such as group size and availability to work.

Table 2 Was the difference in total hours for the plant versus seed method also included in the Total Cost estimate in the second half of the table. Labor is one of the most expensive aspects of restoration and these 190 hours would equate to $4000 plus that should be added to the planting costs. This needs to be revised here and in the discussion.

Response: We understand that labor can contribute significantly to the overall cost of restoration and have thus included the cost of labor in the revised table 2.

The Cost Analysis section has also been revised to better explain this:

“More money was spent restoring with container plants than seed, with the bulk of the costs attributed to labor and supplies needed to grow plants (Table 2). In comparison to simply seeding, we spent 1.4 times more money to use plants that we grew in our own greenhouse and would have spent about 1.2 times more money to use plants purchased from a local nursery. When comparing the cost of perennial seed mixes between restoration techniques (self-grown plants versus seed), more money was spent on perennial seed mixes for the seeding method (Table 2). This is due to the fact the seeding method uses more seed to account for the possibility of loss through drift, predation or inviable seed. Restoration is the most expensive using self-grown plants followed by nursery-grown plants, and lastly seeding (Table 2). Although seeding efforts required a larger volume of seed, the additional cost of supplies and labor associated with nursery-grown plants and self-grown plants resulted in these techniques being more expensive overall.”

The discussion has also been revised to further explain the cost of labor in our experiment:

“Our sowing efforts, which involved hand sown seed followed by raking and tamping, has been shown to be one of the most labor-intensive and time-consuming seeding methods [5]. The combined time and effort associated with growing out seedlings and then planting them in the field was greater than the time it took to simply seed the plots.

When comparing the monetary cost of restoration between restoration techniques, planting was much more expensive than seeding. Restoration efforts using container plants will generally be more expensive due to the added cost of supplies and labor associated with growing and tending to the plants until it is time for planting at the restoration site [12,98]. However, if container plants are necessary for successful restoration then we found that nursery-grown plants may be a cheaper alternative to plants grown at your own facility since the labor costs associated with caring for the plants would be outsourced to nursery staff.

382 The descriptions here of the methods are confusing. Why are you evaluating methods that you did not use? This could be revised to be more clear or reduced for simplicity.

Response: While we did not use nursery grown plants in our study, we still believe it is important to consider the cost of restoration using nursery grown plants in our cost analysis because it is a method that is used by local land managers. We think it is helpful to evaluate the cost difference between restoration efforts using seed, self-grown plants and nursery grown plants in order to inform restoration practitioners which method is most cost-effective.

The description of Table 2 has been simplified to say:

“This table reports the total amount of hours and money spent on restoration using seed, self-grown plants, and nursery-grown plants during the first year of the study (2016-2017).”

And the Cost Comparison section has been edited to include the following:

“While we did not use nursery-grown container plants in our study, we included this cost in our evaluation because nursery-grown container plants are often purchased and used in local restoration efforts. In our analysis, we compared the cost of restoration using seed, self-grown plants and nursery-grown plants to determine which method costs the least amount of money.”

390 Why are we talking about planted versus seeded forb treatments at all? They were separate plots that were both seeded, one just happened to be closer to the planted shrubs. This is really confusing.

Response: Thank you for bringing up this point. We understand that it does not make sense to compare the cost of seeded versus planted forbs since forbs were always seeded in our study. Therefore, this sentence has been deleted, and we decided to exclude the cost of restoring forbs from our analysis.

450 Throughout the paper I felt that I needed to know the seeding rate of each species (# seeds not mass) and the viability of each species. How did you determine the seeding rate for each species? Please add this information to S1 Table. It may be that the Elymus seeds had low viability and you can rule that out if you did a seed viability trial for each species. If you didn’t do this, it is OK but it limits your ability to interpret the data.

Response: The table has been edited to include the seeding rate is seeds/m2. We have also included PLS in this table. The description for the table includes the statement “Information on other restoration projects in the area were used to determine seeding rates.”

We believe that viability was not an issue, since seeds from the same lot were able to grow in the greenhouse. We have added the following sentence to the paragraph:

“However, all of the species germinated well in the greenhouse, and we were able to grow out container plants from the same seed lots.”

466 If you add the dollar amount of the time you will see that outplanting was much more costly.

Response: Thank you for your suggestion. We have included the cost of labor in our analysis and indeed see that planting is much more expensive.

Fig 1 add scale bar to plot design, make larger map smaller if possible

Response: Fig 1 (the site map) has been removed, as it added little to the manuscript.

S1 Table – Please put seeding rate in # seeds/sq m for easier comparison with other studies; include viability for each species and how the seeding rate for each species was determined.

Response: The table has been edited to include the seeding rate is seeds/m2. We have also included PLS in this table. The description for the table includes the statement “Information on other restoration projects in the area were used to determine seeding rates.”

---

## [Decision Letter · Decision Letter 1]

29 Jun 2021

PONE-D-20-35684R1

Is it best to add native shrubs to a coastal sage scrub restoration project as seeds or as seedlings?

PLOS ONE

Dear Dr. McGuire,

Thank you for submitting your manuscript to PLOS ONE. After careful consideration, we feel that it has merit but does not fully meet PLOS ONE’s publication criteria as it currently stands. Therefore, we invite you to submit a revised version of the manuscript that addresses the points raised during the review process.

ACADEMIC EDITOR: Thank you for your through review. Please address reviewer and my relative minor concerns (below) and please pay special attention to formatting details and general editions. Note that PLoS may not send proofs once the paper is accepted for publication.

We look forward to receiving your revised manuscript.

Kind regards,

Cristina Armas

Academic Editor

PLOS ONE

Journal Requirements:

Additional Editor Comments (if provided):

Comments besides those from reviewer:

P5 Please format Latin-scientific names. They should appear in italics.

P5 L. 133 Seeds of some species….

P6L137, L139 and other places. Please avoid one-sentence paragraphs

P6 It is still not clear to me how seeding/sowing was performed within each shrub plot. Did you mixed seeds of all species and sowed them randomly? Was it in a spatial regular pattern to reduce competition (as with planted seedlings)? What about grasses and forbs?

It is neither clear the reasons for including such different number of seeds/m2 per species in the shrub mix plots. Please add some references that support this particular selection.

Please describe with more detail your sowing/seeding method and the reasons for including those different seeding rates per species. This is of particular importance as inn the discussion it is included that (L365) “This, as well as the greater mortality of seeded plants, could be due to high seeding rates. Competition of many emerging plants packed closely together may have limited plant growth and lowered growth and survival in the seeded treatment [83]. Determining seeding rates that are high enough to fill open space without leading to decreased growth due to competitive interactions is one of the complexities of conducting restoration from seed [24].”

P6L152 “to decrease plant competition” Otherwise include a reference that backs up the fact that at those distances among planted individuals there is no plant competition.

Table 2. I would suggest highlighting in bold those treatments that render significant results (this is more common that highlighting in italics those that are not significant).

Reviewers' comments:

Reviewer's Responses to Questions

**Comments to the Author**

1. If the authors have adequately addressed your comments raised in a previous round of review and you feel that this manuscript is now acceptable for publication, you may indicate that here to bypass the “Comments to the Author” section, enter your conflict of interest statement in the “Confidential to Editor” section, and submit your "Accept" recommendation.

Reviewer #2: (No Response)

2. Is the manuscript technically sound, and do the data support the conclusions?

Reviewer #2: Partly

3. Has the statistical analysis been performed appropriately and rigorously? 

Reviewer #2: Yes

4. Have the authors made all data underlying the findings in their manuscript fully available?

Reviewer #2: Yes

5. Is the manuscript presented in an intelligible fashion and written in standard English?

Reviewer #2: Yes

6. Review Comments to the Author

Reviewer #2: This revision addresses most of the concerns from the prior review; however, several new issues need attention. One major issue that the authors should address is that the species that drove the patterns in the seeded treatment (Acmispon glaber) was seeded at a much higher seeding rate than all other species (over 2000 seeds/m2 compared to less than 100 seeds for some other species). Because of this, the results really just show the effect of the treatments on this species. I think a caveat explaining this should be added to the discussion, and some justification for the discrepancy in seeding rates among species should be provided.

68 Correct typo “success restoration success”

74 The word “harsh” is subjective. Consider removing.

89 “We hypothesized that container plants would be best for increasing cover, while seeds would be best for increasing richness and that the plants grown from seed might grow faster and have higher survivorship due to improved root health.” These hypotheses seem to come out of the blue here. They need to be introduced earlier in the introduction. It is not obvious why container plants would be best for cover, but seeds would be best for richness. Is it because you can also add annuals as seed? Why would plants grown as seed have healthier roots than container plants? This was not introduced. Please provide some background to support the hypotheses.

100, 102, etc. On line 100 South is capitalized. On line 102 north is not capitalized. Please be consistent throughout the paper.

137-140 Combine these stray sentences into one of the paragraphs.

150-151 “Between 47 and 69 of each Stipa pulchra and Stipa lepida individuals were planted in their respective plots.” Does this mean 47 S. pulchra and 69 S. lepida? Or were the numbers variable within each species? Please revise for clarity.

151 Provide a reference for the planting density to show there was no competition. Otherwise you should not infer there was no competition without measuring it in the study.

156 I think you can remove the heading “Maintenance” as this is just one sentence that can be included at the end of the prior section.

220 List all the factors and interaction terms for each model. Otherwise it is not clear. Did the repeated measures model also include fixed factors? Which interaction terms were tested in each model? If no interaction terms were included why was this decision made? The logistic regression model does not list any factors at all. It seems that species x treatment should be included in all models with species as a factor since that was one of the reasons for the experimental design. Perhaps referring to Table 1 which shows the fixed factors for each model would be sufficient.

Table 1 – The information in this table is helpful for interpreting the results; however, the formatting as one large table is kind of unusual. I suggest looking in the journal for other examples and formatting the table in a similar way. I also suggest reducing the number of decimal places to make it easier to read.

Results – When an interaction term is significant it is best to interpret only the interaction term and not the individual factors. For example, if species x treatment is significant tell us which species were greater in seeded and which were greater in planted and which showed no effect of treatment, but there is no need to also discuss the overall differences among seeded versus planted. This approach should be applied to cover of native shrubs and lifespan.

Table 3 This table should indicate that the costs for nursery-grown plants are estimated since they were not used in this study.

363 Competition is one possibility for higher mortality, but so is desiccation stress during establishment. Seedlings in the seeded plots were smaller with less developed root systems and may not have tolerated stress as well as the planted shrubs. This is a pretty well-established barrier to restoration in most systems and is a reason why planted shrubs are used.

7. PLOS authors have the option to publish the peer review history of their article (what does this mean?). If published, this will include your full peer review and any attached files.

Reviewer #2: No

---

## [Author Response · Author response to Decision Letter 1]

13 Aug 2021

RESPONSE TO ACADEMIC EDITOR

Response: To our knowledge, none of the references cited have been retracted.

P5 Please format Latin-scientific names. They should appear in italics.

Response: Thank you for catching this. The formatting has been corrected.

P5 L. 133 Seeds of some species….

Response: This sentence has been edited. It now reads:

“Seeds of some species (Acmispon glaber, Eschscholzia californica, Lupinus bicolor, Lupinus succulentus, Malosma laurina, Penstemon spectabilis, Phacelia cicutaria, and Rhus integrifolia) were subjected to dormancy-breaking techniques prior to seeding (S1 Table).”

P6L137, L139 and other places. Please avoid one-sentence paragraphs

Response: These single-sentences have been combined into a paragraph. It reads:

“In January 2017, all experimental plots were seeded and planted according to the randomized plot design. For seeding, we used hand-broadcasting, raking, and tamping, which has been shown to yield greater germination in coastal sage scrub communities compared to other methods [38].”

P6 It is still not clear to me how seeding/sowing was performed within each shrub plot. Did you mixed seeds of all species and sowed them randomly? Was it in a spatial regular pattern to reduce competition (as with planted seedlings)? What about grasses and forbs?

It is neither clear the reasons for including such different number of seeds/m2 per species in the shrub mix plots. Please add some references that support this particular selection.

Please describe with more detail your sowing/seeding method and the reasons for including those different seeding rates per species. This is of particular importance as inn the discussion it is included that (L365) “This, as well as the greater mortality of seeded plants, could be due to high seeding rates. Competition of many emerging plants packed closely together may have limited plant growth and lowered growth and survival in the seeded treatment [83]. Determining seeding rates that are high enough to fill open space without leading to decreased growth due to competitive interactions is one of the complexities of conducting restoration from seed [24].”

Response: The following sentences have been added to further specify the seeding technique:

“Seeds of all shrub, grass, and forb species were combined into their respective seed mixes. The seeding rates varied greatly between different species, as is common in coastal sage scrub restoration [36,37]. Due to previous difficulties with restoring the site, we chose to use higher seeding rates than are typical. The seeding rate for Acmispon glaber is unusually high due in part to a calculation error, in which the intended seeding rate was doubled.”

“Seed mixes were sprinkled onto the ground by hand, using no specific spatial pattern, while taking care to distribute the seed evenly throughout each plot.”

To address the differences in seeding rates for the different species, the following references have been added:

Aprahamian AM, Lulow ME, Major MR, Balazs KR, Treseder KK, Maltz MR. Arbuscular mycorrhizal inoculation in coastal sage scrub restoration. Botany. 2016 Jun;94(6):493–9.

Tamura N, Lulow ME, Halsch CA, Major MR, Balazs KR, Austin P, et al. Effectiveness of seed sowing techniques for sloped restoration sites. Restoration Ecology. 2017;25(6):942–52.

P6L152 “to decrease plant competition” Otherwise include a reference that backs up the fact that at those distances among planted individuals there is no plant competition.

Response: This sentence has been rephrased. It now reads:

“We spaced out all the plants within their plot boundaries to decrease plant competition.”

Table 2. I would suggest highlighting in bold those treatments that render significant results (this is more common that highlighting in italics those that are not significant).

Response: Thank you for the suggestion. Significant results are now shown in bold.

 

RESPONSE TO REVIEWER 2

Reviewer #2: This revision addresses most of the concerns from the prior review; however, several new issues need attention. One major issue that the authors should address is that the species that drove the patterns in the seeded treatment (Acmispon glaber) was seeded at a much higher seeding rate than all other species (over 2000 seeds/m2 compared to less than 100 seeds for some other species). Because of this, the results really just show the effect of the treatments on this species. I think a caveat explaining this should be added to the discussion, and some justification for the discrepancy in seeding rates among species should be provided.

This is a very valid point. We reviewed our original seeding rate calculations for the study and found the cause of this discrepancy. In the Materials and Methods section, we explain that the seeding rate for all species are intentionally higher than usual due to previous difficulties restoring the site. We also explain that the rate for Acmispon glaber “is unusually high due in part to a calculation error, in which the intended seeding rate was doubled.”

The Discussion section has been revised to note this as well. It explains:

“Seeded plots outperformed planted plots in terms of native shrub density, but this trend was driven by one species, Acmispon glaber.”

and 

“The two species that established best from seed in our study, Atriplex lentiformis and Acmispon glaber, are both drought-deciduous at our site, and they are both known to colonize areas from seed following wildfires [90,91]. A. glaber had a greater density and percent cover than any other species in the seeded treatment, likely due to its unintentionally high seeding rate. This species drove the trends in seeded plot cover and density. Its role in the coastal sage scrub community as an important early-successional fire-follower [92] and as a species that grows quickly in recently cleared shrublands [93] may have also contributed to A. glaber’s success when seeded. Further studies could investigate this possibility.”

To address the differences in seeding rates for the different species, references have been added. These show that using greatly different seeding rates for coastal sage scrub species is common in restoration projects.

Aprahamian AM, Lulow ME, Major MR, Balazs KR, Treseder KK, Maltz MR. Arbuscular mycorrhizal inoculation in coastal sage scrub restoration. Botany. 2016 Jun;94(6):493–9.

Tamura N, Lulow ME, Halsch CA, Major MR, Balazs KR, Austin P, et al. Effectiveness of seed sowing techniques for sloped restoration sites. Restoration Ecology. 2017;25(6):942–52.

68 Correct typo “success restoration success”

Response: This typo has been corrected.

74 The word “harsh” is subjective. Consider removing.

Response: As suggested, the word “harsh” has been removed. The sentence now reads:

“For one, using container plants is generally much more expensive than purchasing seeds [13], which places limits on the quality of restoration that can be achieved with a given budget.”

89 “We hypothesized that container plants would be best for increasing cover, while seeds would be best for increasing richness and that the plants grown from seed might grow faster and have higher survivorship due to improved root health.” These hypotheses seem to come out of the blue here. They need to be introduced earlier in the introduction. It is not obvious why container plants would be best for cover, but seeds would be best for richness. Is it because you can also add annuals as seed? Why would plants grown as seed have healthier roots than container plants? This was not introduced. Please provide some background to support the hypotheses.

Response: This section has been revised to better explain our reasoning. It reads:

“We hypothesized that container plants would be best for increasing cover because they are larger when added to the site, while seeds are generally better for increasing richness because they allow for annuals to be added. We hypothesized that seeded plots would have more unique species growing in a standardized area due to smaller stature of seedlings that germinated in the field. Plants grown from seed might grow faster and have higher survivorship due to improved root health [32,33]. We also hypothesized that restoration from seed would be less expensive and time-consuming than restoration from container plants [5]. Finally, we did not have any reason to expect the plants’ success to vary significantly by species or functional traits.”

100, 102, etc. On line 100 South is capitalized. On line 102 north is not capitalized. Please be consistent throughout the paper.

Response: All instances of cardinal directions have been decapitalized.

137-140 Combine these stray sentences into one of the paragraphs.

Response: These single-sentences have been combined into a paragraph. It reads:

“In January 2017, all experimental plots were seeded and planted according to the randomized plot design. For seeding, we used hand-broadcasting, raking, and tamping, which has been shown to yield greater germination in coastal sage scrub communities compared to other methods [38].”

150-151 “Between 47 and 69 of each Stipa pulchra and Stipa lepida individuals were planted in their respective plots.” Does this mean 47 S. pulchra and 69 S. lepida? Or were the numbers variable within each species? Please revise for clarity.

Response: To improve clarity, the sentence now reads:

“Between 47 and 69 Stipa individuals were added to each planted-treatment grass plot.”

151 Provide a reference for the planting density to show there was no competition. Otherwise you should not infer there was no competition without measuring it in the study.

Response: This sentence has been rephrased. It now reads:

“We spaced out all the plants within their plot boundaries to decrease plant competition.”

156 I think you can remove the heading “Maintenance” as this is just one sentence that can be included at the end of the prior section.

Response: The heading “Maintenance” has been removed.

220 List all the factors and interaction terms for each model. Otherwise it is not clear. Did the repeated measures model also include fixed factors? Which interaction terms were tested in each model? If no interaction terms were included why was this decision made? The logistic regression model does not list any factors at all. It seems that species x treatment should be included in all models with species as a factor since that was one of the reasons for the experimental design. Perhaps referring to Table 1 which shows the fixed factors for each model would be sufficient.

Response: We agree that this was confusing, and we have revised the data analysis section of the Methods to clarify the factors included in each model, as recommended. In particular, we were careful to include all factors included in each model, and to clarify why interactions were included. Note that species was not included when models were based on more holistic measures of restoration success, such as native plant species richness. 

Table 1 – The information in this table is helpful for interpreting the results; however, the formatting as one large table is kind of unusual. I suggest looking in the journal for other examples and formatting the table in a similar way. I also suggest reducing the number of decimal places to make it easier to read.

Response: This table has been reformatted into three sections - one for ANOVA, one for Repeated-Measures ANOVA, and one for Logistic Regression. To improve readability, cells containing the same dependent variable have been merged. When possible values were rounded to three decimal places.

We believe that the use of a large table to report the statistical results is appropriate. We referenced several other PLOS ONE articles to match the formatting as best as possible.

Similar tables are present in the following PLOS articles:

• Changes in Soil Carbon and Nitrogen following Land Abandonment of Farmland on the Loess Plateau, China

• Seasonal Dynamics of the Plant Community and Soil Seed Bank along a Successional Gradient in a Subalpine Meadow on the Tibetan Plateau

• Effects of 10-Year Management Regimes on the Soil Seed Bank in Saline-Alkaline Grassland

• Restoration management of cattle resting place in mountain grassland

Results – When an interaction term is significant it is best to interpret only the interaction term and not the individual factors. For example, if species x treatment is significant tell us which species were greater in seeded and which were greater in planted and which showed no effect of treatment, but there is no need to also discuss the overall differences among seeded versus planted. This approach should be applied to cover of native shrubs and lifespan.

Response: The suggested edits have been made. These sections read:

“Acmispon glaber, Atriplex lentiformis, Encelia Californica, Eriogonum fasciculatum, and Salvia mellifera had greater cover when seeded. Elymus condensatus, Isocoma menziesii, Malosma laurina, and Rhus integrifolia had greater cover when planted, but very low cover in both treatments. Cover of Artemisia californica, Baccharis emoryi, and Peritoma arborea showed no effect of treatment”

“There was a significant treatment-by-species interaction for shrub lifespan. Acmispon glaber and Atriplex lentiformis lived longer when seeded. Artemisia californica, Baccharis emoryi, Encelia californica, Eriogonum fasciculatum, Malosma laurina, Peritoma arborea, and Salvia mellifera had longer lifespans when planted. The remaining species did not have enough surviving individuals to compare.”

Table 3 This table should indicate that the costs for nursery-grown plants are estimated since they were not used in this study.

Response: The following sentence has been added to the description for Table 3:

“It also estimates the cost of using nursery-grown plants.”

363 Competition is one possibility for higher mortality, but so is desiccation stress during establishment. Seedlings in the seeded plots were smaller with less developed root systems and may not have tolerated stress as well as the planted shrubs. This is a pretty well-established barrier to restoration in most systems and is a reason why planted shrubs are used.

Response: Thank you for bringing up this point. This has now been mentioned in the discussion section:

“The higher mortality of seeded shrubs may also be because they were younger and less developed, and therefore may not have been able to tolerate environmental stress as well as their planted counterparts [87].”

---

## [Editor Report · Decision Letter 2]

27 Dec 2021

Is it best to add native shrubs to a coastal sage scrub restoration project as seeds or as seedlings?

PONE-D-20-35684R2

Dear Dr. McGuire,

We’re pleased to inform you that your manuscript has been judged scientifically suitable for publication and will be formally accepted for publication once it meets all outstanding technical requirements.

Kind regards,

Cristina Armas

Academic Editor

PLOS ONE
---

## [Editor Report · Acceptance letter]

31 Jan 2022

PONE-D-20-35684R2 

Is it best to add native shrubs to a coastal sage scrub restoration project as seeds or as seedlings? 

Dear Dr. McGuire:

I'm pleased to inform you that your manuscript has been deemed suitable for publication in PLOS ONE. Congratulations! Your manuscript is now with our production department. 

Kind regards, 

on behalf of

Dr. Cristina Armas 

%CORR_ED_EDITOR_ROLE%

PLOS ONE